# Symptom improvement and predictors associated with improvement after 6 weeks of alpha-blocker therapy: An exploratory, single-arm, open-label cohort study

Henk van der Worp[1]*, Boudewijn J. Kollen[1], Tom Vermist[1], Martijn G. Steffens[2], Marco H. Blanker[1]

1 Department of General Practice and Elderly Care Medicine, University of Groningen, University Medical Center Groningen, Groningen, The Netherlands, 2 Department of Urology, Isala Clinics Zwolle, Zwolle, The Netherlands

* h.van.der.worp@umcg.nl

**Data Availability Statement:** All relevant data are within the manuscript and its Supporting Information files.

## Abstract

### Objectives

Clinicians should not only know how many patients will benefit from alpha-blocker therapy but should also be able to identify who will benefit. We studied the changes in patient symptoms following alpha-blocker therapy and the predictors of symptom improvement in clinical practice.

### Design

This was a single-arm, open-label observational cohort study with a 6-week follow-up.

### Setting

Twenty-two pharmacies in the Netherlands.

### Participants

Patients were eligible for inclusion if they attended a pharmacy with a new prescription for an alpha-blocker from a general practitioner or urologist.

### Primary and secondary outcomes

Outcomes were assessed using the International Prostate Symptom Score (IPSS), Overactive Bladder Questionnaire Short Form (OAB-q SF), and Patient Global Impression of Improvement (PGI-I). Demographic, disease-related, and drug-related information were collected to identify predictors of symptom improvement. These predictors were then assessed by logistic and linear regression analyses of both the original data set and an imputed data set that accounted for the missing variables.

**Funding:** This work was supported by The Hein Hogerzeil Foundation with an unrestricted grant to MB. The Foundation was not involved in the design of this study, nor in the data collection, data analyses, interpretation of the outcomes, or the writing of this manuscript.

**Competing interests:** The authors have declared that no competing interests exist.

## Results

During the study, 37% of patients with lower urinary tract symptoms perceived clear symptomatic improvement based on the results of the PGI-I. Improvement was more likely in those who still used alpha-blockers at the end of the 6-week study period and in those who used multiple medications. Although symptom scores decreased significantly on the IPSS and OAB-q SF, the only predictor of change was the pretreatment symptom severity.

## Conclusions

Approximately one-third of our cohort perceived symptom improvement on alpha-blocker therapy. However, we identified no clear predictors of who might benefit from alpha-blocker treatment, indicating that alpha-blockers should still be prescribed on a trial basis.

## Introduction

Alpha-blockers are the first step of drug treatment for men with moderate to severe lower urinary tract symptoms (LUTS).[1,2] Current guideline recommendations are based on the efficacy of alpha-blockers assessed by symptom scores in randomized controlled trials.[2–4] In daily practice, however, patients and clinicians are more interested in achieving symptom improvements that reflect changes in the subjective appraisal of symptoms, which may not be reflected on these symptom scores. Predicting the likelihood of improvement is difficult for clinicians because the randomized controlled trials on which they rely for guidance tend to report mean changes in symptom scores rather than the number of patients who experience improvement. Moreover, the inability to determine who will benefit from alpha-blocker treatment is confounded by the multifactorial origin of LUTS in the male population. Clinicians must therefore rely on the general advice to start alpha-blockers when treatment is requested.[3–5] Information that helps clinicians to identify those patients who are most likely to experience improvement after alpha-blocker use would be clinically relevant.

In this study, we aimed to assess symptom improvement following alpha-blocker use and to identify the predictors of that improvement in routine clinical practice.

## Materials and methods

We performed a single-arm, open-label, observational cohort study with a 6-week follow-up from January 2016 to April 2018. The primary settings were 22 pharmacies in the Northern part of the Netherlands. Patients were eligible for inclusion if they attended a pharmacy with a new prescription for an alpha-blocker to treat LUTS, provided they had not been prescribed an alpha-blocker in the preceding 12 months. The following exclusion criteria were also checked by the pharmacy: indwelling urinary catheter use, urolithiasis, and status as a first prescription as part of combination therapy (i.e., alpha-blocker combined with an anticholinergic drug or a 5-alpha reductase inhibitor).

Participants completed a questionnaire at baseline, and we subsequently obtained data from a follow-up questionnaire and from the records of the patient's general practitioner (GP) or urologist, depending on the prescriber, as well as the pharmacist. The requirements of the STROBE statement were followed when reporting the study.[6] The Medical Ethical Committee of the University Medical Center Groningen approved the study (METc 2016.122) and all participants provided signed informed consent before inclusion.

## Patient questionnaire

The baseline questionnaire included questions about age, duration of LUTS, and previous surgery for LUTS. It also included the International Prostate Symptom Score (IPSS) and the Overactive Bladder Questionnaire Short Form (OAB-q SF). The IPSS consists of seven questions that assess urinary symptoms,[7] giving total scores of 0–35, with higher scores indicating worse symptoms. The OAB-q SF assesses how bothersome overactive bladder symptoms are for the patient.[8] It consists of six questions, with total scores ranging from 0 to 100 and higher scores indicating greater bother from symptoms. The follow-up questionnaire included the IPSS, the OAB-q SF, and the Patient Global Impression of Improvement (PGI-I),[9] together with questions about current alpha-blocker use and (if relevant) reasons for discontinuing therapy.

## GP, urologist, and pharmacist questionnaires

GPs and urologists were asked to provide data about comorbidities and physical examinations. Pharmacists were asked to provide a list of concomitant medication, from which we extracted data about the number of drugs currently used. We selected specific drugs with the potential to affect LUTS, using the Anatomical Therapeutic Chemical Classification System for categorization (e.g., antipsychotics, tricyclic antidepressants, selective serotonin reuptake inhibitors, Parkinson's disease medication, classic antihistamines, calcium antagonists, and opiates).[4]

**Outcomes.** The main outcomes were clear symptom improvement, assessed by the PGI-I, as well as the predictors of that improvement. For this we categorized the responses "much better" or "very much better" as *clear improvement* and all other categories as *no clear improvement*. Secondary outcomes were the changes in symptoms, assessed by the IPSS and OAB-q, as well as the predictors of those changes. For this, we compared the responses after 6 weeks of therapy with the baseline scores.

## Handling of missing data

Missing value analysis suggested that data were missing at random for approximately 5% of baseline data and 35% of follow-up data. Multiple imputation was performed on this dataset with 35 imputations and 20 iterations, using fully conditional specification and predictive mean matching. Patient characteristics, symptom scores, and medical information were used in the imputation model. Analyses performed on the imputed datasets were pooled. All analyses were performed on both the original complete data and the imputed data.

## Statistical analyses

Baseline characteristics are presented for the original data. Percentages experiencing clear improvement and the change in symptom severity are presented for both the original and imputed data. We assessed the change in symptoms following alpha-blocker use by comparing baseline and follow-up scores on the IPSS and AOB-q, using paired $t$-tests. All analyses were performed using IBM SPSS Statistics for Windows, Version 25.0 (IBM Corp., Armonk, NY, USA). An alpha of $<0.05$ was considered statistically significant in all analyses, unless otherwise stated.

To explore the possible predictors of clear improvement, we performed univariable logistic regression, setting clear improvement as the dependent variable. All assumptions for the test were met. We included the following variables as potential predictors: IPSS score, IPSS storage sub-score, IPSS voiding sub-score, OAB-q SF score, age category (i.e., <60, 60–70, or >70 years), symptom duration (i.e., <6, 6–24, or >24 months), continued alpha-blocker use,

comorbidities (i.e., yes or no), prescriber (i.e., GP or urologist), prostate abnormalities, pelvic floor abnormalities, number of co-medications (i.e., 0–1, 2–5, or $\geq$6, based on tertiles), and use of medication with a potential effect on LUTS. Variables with a p-value <0.25 in the univariate analyses were included in the multivariable regression analyses by a non-automated step-wise forward selection strategy. In the multivariable analyses, we used ten events per variable to obtain the maximum number of independent variables for inclusion.[10] Model fit was assessed by the Hosmer–Lemeshow test, explained variance by the Nagelkerke $R^2$, and discrimination was assessed by the area under the receiver operating characteristic curve.

Linear regression analyses were performed, with the changes in the IPSS and OAB-q scores between baseline and follow-up set as the outcomes of interest, using the predictors and variable selection procedures described above. Dummy variables were created for categorical variables with two or more categories. The assumptions for the test were met and the maximum number of independent variables that could be included was determined based on accepted guidelines.[11] The explained variance was assessed based on the adjusted $R^2$. Because a higher IPSS score indicates more symptoms, regression coefficients of improvement will have a negative sign. Subgroup regression analyses were performed for participants who had their therapy prescribed by a GP and had complete data.

## Results

Of the 258 screened patients, 251 met the inclusion criteria. One patient did not provide informed consent and two died during follow-up, leaving 248 participants for analysis. Complete data were obtained for 119 cases, and their baseline characteristics are presented in Table 1. An overview of the medication that was used by the patients is given in S1 Table. Most patients reported moderate or severe symptoms, reported that symptoms had been present for over 24 months, and received their prescription from a GP (84%). Although most were still taking therapy at 6 weeks (n = 89; 75%), several had stopped therapy because of ineffectiveness (n = 12), side effects (n = 9), or lack of awareness that the medication should be continued (n = 9).

### Primary outcome: Clear improvement

In both the original and imputed data sets, 37% of participants reported clear symptom improvement. For the original data, currently using alpha-blockers and currently using at least six other medications both predicted clear improvement in both the univariate and multivariate analyses (Table 2). However, we found no associations for any other potential predictor. The imputed data analyses also yielded no significant predictors of clear improvement. The multivariate models were similar for the original and imputed data, with explained variances (Nagelkerke $R^2$) of 20.6% and 10.9%, respectively. Model fit was better when based on the original data (Table 2).

### Secondary outcomes: Change in symptom severity

The mean improvements in the IPSS were 6.3 points (95%CI 5.1–7.6) for the original data and 5.4 points (95%CI 4.0–6.7) for the imputed data. In the multivariate regression, multicollinearity existed between the pretreatment IPSS sub-scores and IPSS total score, so we only included the IPSS total score as a predictor (based on the adjusted $R^2$). Univariate regression analysis using the original data showed that men receiving at least six co-medications had greater symptom improvements; but, this was not confirmed in the multivariate regression analysis (Table 3). Multivariate analyses on both datasets showed that higher baseline symptom severity was associated with greater improvements after 6 weeks, but there were no other independent

**Table 1. Baseline characteristics.**

|  | Complete cases (original data) |
| --- | --- |
|  | **N = 119** |
| Age, (mean ± SD) | 66.3 ± 9.3 y |
| Age category, N (%) |  |
| <60 y | 31 (26.1) |
| 60–70 y | 47 (39.5) |
| >70 y | 41 (34.5) |
| Duration of complaints, N (%) |  |
| ≤6 months | 40 (33.6) |
| >6 months to ≤24 months | 15 (12.6) |
| >24 months | 64 (53.8) |
| IPSS score (mean ± SD) | 19.4 ± 6.8 |
| IPSS category |  |
| Mild | 3 (2.5) |
| Moderate | 61 (51.3) |
| severe | 55 (46.2) |
| Storage sub-score (mean ± SD) | 8.6 ± 3.2 |
| Voiding sub-score (mean ± SD) | 10.8 ± 5.1 |
| OAB-q SF (mean ± SD) | 39.7 ± 19.9 |
| Surgery for LUTS (%) | 1 (0.8) |
| Comorbidity*, N (%) | 35 (29.4) |
| Prescriber N (%) |  |
| GP | 100 (84) |
| Urologist | 19 (16) |
| Prostate abnormal, N (%) |  |
| Normal | 38 (31.9) |
| Increased size | 51 (42.9) |
| Decreased size | 3 (2.5) |
| Not examined | 27 (22.7) |
| Examination of pelvic floor, N (%) |  |
| Hypertonic | 7 (5.9) |
| Not hypertonic | 54 (45.4) |
| Not determined | 34 (28.6) |
| Not examined | 24 (20.2) |
| Number of co-medications, N (%) |  |
| 0–1 | 48 (40.3) |
| 2–5 | 49 (41.2) |
| >5 | 22 (18.5) |
| Co-medication with effect on LUTS, N (%) | 15 (12.6) |

* Comorbidities included diabetes, stroke, myocardial infarction, and heart failure.

*Abbreviations*: IPSS, International Prostate Symptom Score; LUTS, Lower Urinary Tract Symptoms; OAB-q SF, Overactive Bladder Questionnaire Short Form; SD, standard deviation.

predictors of change in IPSS scores. We included prostate abnormalities in the model using imputed data, but not in the model using original data. The percentages of variance explained by the original and imputed data were 30.0% and 33.5%, respectively.

Finally, the OAB-q SF score improved by 12.0 points (95%CI 8.8–15.2) in the original data and by 11.7 points (95%CI 8.2–15.1) in the imputed data. In both datasets, only pretreatment symptom scores were significant predictors of change in the OAB-q in the univariate regression analyses (Table 4). Greater baseline symptom severity was associated with greater symptom improvements. In the multiple regression analyses, the OAB-q SF pretreatment score remained the only predictor of improvement. The multivariate model using original data included current alpha-blocker use and the model using imputed data included the presence of comorbidities. The percentages of variance explained by the original and imputed data were 28.7% and 36.0%, respectively.

## Subgroup analyses

Analyses on participants who received their first prescription from a GP (n = 100; 84%) showed similar results to those of the total group (S2–S4 Tables). Here also 37% of participants

**Table 2. Predictors of clear PGI-I improvement.**

| | Original data (complete cases) | | | | Imputed data (pooled outcomes) | | | |
| --- | --- | --- | --- | --- | --- | --- | --- | --- |
| | Univariable analysis | | Multivariable analysis | | Univariable analysis | | Multivariable analysis | |
| | OR | 95% CI | OR | 95% CI | OR | 95% CI | OR | 95% CI |
| Age (ref = <60 y) | | | | | | | | |
| 60–70 y | 1.70 | (0.66;4.38) | NI | | 1.23 | (0.55;2.71) | NI | |
| >70 y | 0.98 | (0.36;2.65) | NI | | 0.98 | (0.45;2.14) | NI | |
| Duration of complaints (ref = <6 months) | | | | | | | | |
| 6–24 months | 0.55 | (0.16;1.91) | 0.61 | (0.16;2.34) | 0.58 | (0.21;1.62) | 0.59 | (0.20;1.80) |
| > 24 months | 0.50 | (0.22;1.14) | 0.49 | (0.20;1.18) | 0.64 | (0.31;1.32) | 0.66 | (0.31;1.40) |
| IPSS sum score | 0.99 | (0.94;1.05) | NI | | 0.99 | (0.94;1.04) | NI | |
| IPSS storage baseline | 0.98 | (0.88;1.10) | NI | | 0.96 | (0.87;1.06) | NI | |
| IPSS voiding baseline | 1.00 | (0.93;1.07) | NI | | 0.99 | (0.93;1.06) | NI | |
| OAB-q SF baseline | 1.00 | (0.98;1.02) | NI | | 1.00 | (0.99;1.02) | NI | |
| Still using α-blockers at 6 weeks (ref = no) | **5.31** | **(1.71;16.47)** | **5.70** | **(1.73;18.84)** | 2.21 | (0.75;6.52) | 2.42 | (0.80;7.30) |
| Comorbidity (ref = no) | 1.20 | (0.53;2.70) | NI | | 1.29 | (0.55;2.98) | NI | |
| Prescriber (ref = GP) | 0.99 | (0.36;2.75) | NI | | 0.98 | (0.41;2.33) | NI | |
| Prostate abnormal (ref = no) | | | | | | | | |
| Increased size | 0.94 | (0.39;2.24) | NI | | 0.95 | (0.43;2.08) | NI | |
| Decreased size | 3.43 | (0.28;41.32) | NI | | 1.27 | (0.18;9.01) | NI | |
| Not examined | 1.01 | (0.36;2.80) | NI | | 0.92 | (0.32;2.64) | NI | |
| Examination of pelvic floor (ref = hypertonic) | | | | | | | | |
| Not hypertonic | 4.13 | (0.46;36.69) | NI | | 1.36 | (0.17;11.01) | NI | |
| Not determined | 3.27 | (0.35;30.46) | NI | | 1.36 | (0.27;9.97) | NI | |
| Not examined | 3.60 | (0.37;34.93) | NI | | 1.19 | (0.19;10.24) | NI | |
| Number of co-medication (ref = 0–1) | | | | | | | | |
| 2–5 | 1.56 | (0.66;3.70) | 1.27 | (0.51;3.17) | 1.13 | (0.38;3.31) | 1.03 | (0.32;3.29) |
| ≥6 | **3.89** | **(1.35;11.25)** | **3.80** | **(1.23;11.70)** | 2.19 | (0.80;5.96) | 2.28 | (0.76;6.82) |
| Co-medication with an effect on LUTS (ref = no) | 0.83 | (0.27;2.62) | NI | | 1.62 | (0.46–5.75) | NI | |
| | | | Nagelkerke R² = 20.6% HL test = 0.87 AUC = 0.73 | | | | Nagelkerke R² = 10.9% HL test = 0.77 AUC = 0.66 | |

*Abbreviations*: AUC, Area Under the Curve; CI, confidence interval; GP, General Practitioner; HL, Hosmer–Lemeshow; IPSS, International Prostate Symptom Score; LUTS, Lower Urinary Tract Symptoms; NI, Not included; OAB-q SF, Overactive Bladder Questionnaire Short Form; OR, odds ratio; PGI-I, Patient Global Impression of Improvement.

**Table 3. Predictors of IPSS improvement.**

| | Original data (complete cases) | | | | Imputed data (pooled outcomes) | | | |
|---|---|---|---|---|---|---|---|---|
| | Univariable analysis | | Multivariable analysis | | Univariable analysis | | Multivariable analysis | |
| | IPSS change | 95% CI | IPSS change | 95% CI | IPSS change | 95% CI | IPSS change | 95% CI |
| Constant | | | 0.28 | (-4.58;5.14) | | | 3.22 | (-1.31;7.75) |
| Age (ref = <60 y) | | | | | | | | |
| *60–70 y* | 1.13 | (-2.09;4.35) | NI | | 0.92 | (-1.59;3.43) | NI | |
| *>70 y* | 0.82 | (-2.49;4.14) | NI | | 1.40 | (-1.22;4.02) | NI | |
| Duration of complaints (ref = <6 months) | | | | | | | | |
| *6–24 months* | 1.04 | (-3.18;5.26) | NI | | 0.18 | (-3.02;3.37) | NI | |
| *>24 months* | 0.20 | (-2.61;3.01) | NI | | -0.39 | (-2.77;1.99) | NI | |
| IPSS sum score | **-0.54** | **(-0.70;-0.39)** | **-0.59** | **(-0.79;-0.39)** | **-0.61** | **(-0.74;-0.47)** | **-0.63** | **(-0.79;-0.48)** |
| IPSS storage baseline | **-0.77** | **(-1.14;-0.40)** | NI* | | **-0.82** | **(-1.13;-0.51)** | NI* | |
| IPSS voiding baseline | **-0.67** | **(-0.89;-0.45)** | NI* | | **-0.73** | **(-0.91;-0.54)** | NI* | |
| OAB-q SF baseline | **-0.06** | **(-0.13;-0.01)** | 0.06 | (-0.01;0.13) | **-0.09** | **(-0.14;-0.04)** | 0.04 | (-0.02;0.09) |
| Still using α-blockers at 6 weeks (ref = no) | -1.03 | (-3.96;1.90) | NI | | -0.70 | (-3.99;2.59) | NI | |
| Comorbidity (ref = no) | -1.99 | (-4.76;0.78) | -0.12 | (-2.96;2.72) | 0.11 | (-2.61;2.84) | NI | |
| Prescriber (ref = GP) | 2.91 | (-0.53;6.34) | 2.42 | (-0.55;5.38) | 1.92 | (-0.79;4.62) | 1.28 | (-1.10;3.66) |
| Prostate abnormal (ref = no) | | | | | | | | |
| *Increased size* | 0.33 | (-2.66;3.31) | NI | | 0.28 | (-2.94;3.49) | 0.24 | (-2.36;2.83) |
| *Decreased size* | 4.40 | (-3.96;12.77) | NI | | 2.94 | (-2.91;8.79) | 1.47 | (-3.66;6.60) |
| *Not examined* | 0.66 | (-2.84;4.17) | NI | | 0.42 | (-3.06;3.90) | 0.27 | (-2.55;3.08) |
| Examination of pelvic floor (ref = hypertonic) | | | | | | | | |
| *Not hypertonic* | 1.01 | (-4.60;6.61) | NI | | -0.95 | (-5.68;3.77) | NI | |
| *Not determined* | 2.09 | (-3.70;7.88) | NI | | 0.27 | (-5.25;5.80) | NI | |
| *Not examined* | 2.32 | (-3.68;8.31) | NI | | 0.33 | (-5.17;5.83) | NI | |
| Number of co-medications (ref = 0–1) | | | | | | | | |
| *2–5* | -0.43 | (-3.13;2.28) | 0.19 | (-2.42;2.80) | -0.60 | (-3.52;2.32) | -0.08 | (-2.15;2.31) |
| *≥6* | **-5.60** | **(-9.03;-2.18)** | -2.20 | (-6.03;1.62) | -3.30 | (-7.89;1.28) | -1.05 | (-4.09;1.99) |
| Co-medication with an effect on LUTS (ref = no) | 0.92 | (-2.92;4.75) | NI | | 2.00 | (-1.69;5.69) | NI | |
| | | | Adjusted $R^2$ = 30.0% | | | | Adjusted $R^2$ = 33.5% | |

Negative regression coefficients indicate improvement of symptoms.

*NI due to multicollinearity.

*Abbreviations*: CI, confidence interval; GP, General Practitioner; IPSS, International Prostate Symptom Score; NI, Not included; OAB-q SF, Overactive Bladder Questionnaire Short Form.

reported clear symptom improvement. In this subgroup patients with symptoms for over 24 months were less likely to show clear improvement compared to those with symptoms for less than 6 months (OR 0.36 95% CI 0.13–0.98).

## Discussion

The PGI-I results showed that about one in three men with LUTS perceived clear symptom improvement by 6 weeks of starting an alpha-blocker, implying that two in three men

**Table 4. Predictors of OAB-q SF improvement.**

| | Original data (complete cases) | | | | Imputed data (pooled outcomes) | | | |
| --- | --- | --- | --- | --- | --- | --- | --- | --- |
| | Univariable analysis | | Multivariable analysis | | Univariable analysis | | Multivariable analysis | |
| | OAB-q SF change | 95% CI | OAB-q SF change | 95% CI | OAB-q SF change | 95% CI | OAB-q SF change | 95% CI |
| Constant | | | 8.53 | (-1.00;18.06) | | | 8.80 | (0.43;17.17) |
| Age (ref = <60 y) | | | | | | | | |
| *60–70 y* | 3.59 | (-4.46;11.63) | NI | | 2.96 | (-3.82;9.75) | NI | |
| *>70 y* | 2.16 | (-6.11;10.44) | NI | | 3.32 | (-3.99;10.62) | NI | |
| Duration of complaints (ref = <6 months) | | | | | | | | |
| *6–24 months* | 3.63 | (-6.90;14.16) | NI | | 4.28 | (-5.01;13.56) | NI | |
| *>24 months* | 2.68 | (-4.33;9.69) | NI | | 3.62 | (-3.01;10.24) | NI | |
| IPSS sum score | **-0.74** | **(-1.19;-0.30)** | 0.18 | (-0.33;0.68) | **-0.74** | **(-1.17;-0.30)** | 0.23 | (-0.23;0.68) |
| IPSS storage baseline | **-1.26** | **(-2.22;-0.31)** | NI* | | **-1.31** | **(-2.24;-0.37)** | NI* | |
| IPSS voiding baseline | **-0.83** | **(-1.44;-0.22)** | NI* | | **-0.75** | **(-1.30;-0.20)** | NI* | |
| OAB-q SF baseline | **-0.49** | **(-0.62;-0.35)** | **-0.50** | **(-0.66;-0.33)** | **-0.58** | **(-0.72;-0.45)** | **-0.61** | **(-0.77;-0.45)** |
| Still using α-blockers at 6 weeks (ref = no) | -4.46 | (-11.74;2.83) | -3.06 | (-9.30;3.18) | -4.17 | (-11.35;3.01) | | |
| Comorbidity (ref = no) | 0.49 | (-6.50;7.47) | NI | | 5.52 | (-2.32;13.36) | 6.82 | (-1.29;14.93) |
| Prescriber (ref = GP) | 4.57 | (-4.08;13.22) | NI | | 3.65 | (-4.09;11.40) | NI | |
| Prostate abnormal (ref = no) | | | | | | | | |
| *Increased size* | 3.45 | (-4.00;10.90) | NI | | 1.28 | (-6.77;9.32) | NI | |
| *Decreased size* | 11.90 | (-8.95;32.75) | NI | | 4.27 | (-11.65;20.18) | NI | |
| *Not examined* | 2.02 | (-6.73;10.77) | NI | | -0.45 | (-10.13;9.23) | NI | |
| Examination of pelvic floor (ref = hypertonic) | | | | | | | | |
| *Not hypertonic* | -2.02 | (-15.92;11.88) | NI | | -3.67 | (-16.63;9.29) | NI | |
| *Not determined* | 4.52 | (-9.84;18.88) | NI | | -0.04 | (-13.61;13.53) | NI | |
| *Not examined* | 0.26 | (-14.60;15.12) | NI | | -1.82 | (-15.96;12.33) | NI | |
| Number of co-medications (ref = 0–1) | | | | | | | | |
| *2–5* | -4.48 | (-11.44;2.49) | -1.94 | (-7.98;4.10) | -3.42 | (-11.27;4.43) | -3.46 | (-10.07;3.14) |
| *≥6* | -8.51 | (-17.34;0.32) | -3.51 | (-11.68;4.66) | -5.76 | (-15.00;3.48) | -5.33 | (-13.94;3.28) |
| Co-medication with an effect on LUTS (ref = no) | -6.87 | (-16.37;2.64) | -3.18 | (-11.43;5.07) | 1.20 | (-7.44;9.84) | NI | |
| | | | Adjusted R² = 28.7% | | | | Adjusted R² = 36.0% | |

Negative regression coefficients indicate improvement of symptoms.

*NI due to multicollinearity.

*Abbreviations*: CI, confidence interval; GP, General Practitioner; IPSS, International Prostate Symptom Score; LUTS, Lower Urinary Tract Symptoms; NI, Not included; OAB-q SF, Overactive Bladder Questionnaire Short Form.

experienced no clear improvement. Current use of an alpha-blocker and current use of at least six additional medications predicted clear improvement, but only in the model using original data. Although symptoms measured with the IPSS and OAB-q SF decreased significantly by 6 weeks, only pretreatment symptom severity predicted this change. Measures of explained variance for these models suggested that factors not assessed in this study predicted the observed symptom improvement following alpha-blocker use. We believe that this information is relevant to pretreatment counseling.

Although currently use of alpha-blockers and six or more medications were both significant predictors of clear improvement in the analysis of original data, they did not remain predictors in the analysis of imputed data, despite there being similar trends in the odds ratios. The increased odds indicated that clear improvement was more likely for those participants who still used alpha-blockers after 6 weeks. However, no such relationship existed between current alpha-blocker use and the more quantitative measures of improvement (i.e., the IPSS and OAB-q SF), suggesting that any observed improvements in symptoms could not be attributed to changes in either the frequency of urination or episodes of incontinence.

It is difficult to explain why participants who took more medications were more likely to experience clear improvement. One explanation could be that participants who are used to taking medication are more likely to include it in their regimens, improving the effectiveness of alpha-blocker therapy. Supporting this, it has been shown that drug adherence is related to concomitant use of other medications.[12–14] The present study did not monitor actual intake, however, so we cannot comment with certainty. Another explanation is that other types of medication lead to improvement of LUTS, however this is less likely. First of all, other medication was initiated before baseline, and remained unchanged during the observation period. Secondly, in our analyses we looked at known specific drugs with the potential to affect LUTS and found no relation with change in symptoms, although we cannot rule out that there are other types of drugs that have an influence on LUTS.

Only pretreatment scores predicted improvements in the IPSS and OAB-q SF, with the data indicating that participants with more severe symptoms had greater benefit from alpha-blocker treatment. Alternatively, we must consider that this may have been caused by a ceiling effect or regression to the mean, which would have a greater influence in participants with higher baseline scores. No other predictors could be identified. Nevertheless, our findings regarding pretreatment scores are consistent with those of similar studies in secondary care, [15,16] which additionally showed that prostate volume predicted outcomes. Another study produced contradictory findings that there was no relationship,[17] instead proposing that intravesical protrusion of the prostate predicted improvement, as reported elsewhere.[18] However, we restricted our data collection to those characteristics that are readily available to GPs, and neither the prostate volume nor the intravesical prostatic protrusion is routinely measured in general practice.

In many countries, men with LUTS typically consult their GPs first, yet we could find no studies referring to symptom improvement following alpha-blocker use in the primary care setting. We therefore included participants who received their first prescription from either a GP (n = 100; 84%) or a urologist (n = 19; 16%). In the subgroup analysis of participants who were prescribed alpha-blockers by a GP, the results were comparable to those for the total group. Unfortunately, the subgroup of participants receiving prescriptions from a urologist was too small for separate analysis or comparison.

Finally, we showed that 70% of participants had persevered with their treatment by the end of the study. This finding is consistent with that reported in another study, in which 73% persevered with treatment beyond 6 weeks.[19] Other studies have also indicated that approximately 35%–40% persist with therapy beyond 12 months, indicating that discontinuation is most likely in the first few weeks or months after initiation.[19,20] Future studies should further assess the reasons for persistence with alpha-blockers.

A major limitation of this study is the absence of a control group. Another limitation is that the actual intake of medication was not monitored, which makes it impossible to know the true effect of compliance with alpha-blocker therapy. Nevertheless, both results must be interpreted in the context that the study was designed to reflect real-life settings. Only including participants who visited a pharmacy may also have led to selective inclusion by inadvertently

excluding housebound participants who were unable to visit the pharmacy. This small population is likely to be older and to have more severe symptoms than our sample, limiting the generalizability of our results. A final limitation is that there was a significant amount of missing data, with follow-up data missing for 35% and incomplete data available for 50%. Although multiple imputation for the missing values allowed us to perform analyses on the complete sample, the results of analyses were comparable for both data sets, albeit with small differences found between factors included in the multivariate model.

In conclusion, slightly more than one-third of our cohort perceived clear improvement after alpha-blocker therapy. Clinicians can use this information when counseling patients about the risks and benefits of therapy. Overall, however, the study provides little additional information to guide clinicians on who will benefit most from alpha-blocker treatment. As such, advice in existing guidelines should continue to be followed, with alpha-blocker therapy trialed for all men with moderate to severe LUTS who request treatment.

## Supporting information

**S1 Dataset. Minimal dataset.**
(SAV)

**S1 Table. Overview of the most used types of medication grouped on ATC-code.**
(DOCX)

**S2 Table. Analyses on patients with a prescription from their GP and complete original data (n = 100). Predictors of clear improvement in PGI-I.** AUC = Area Under the Curve; NI = Not included.
(DOCX)

**S3 Table. Analyses on patients with a prescription from their GP and complete original data (n = 100). Predictors of improvement in IPSS.** NI = Not included; *NI due to multicollinearity.
(DOCX)

**S4 Table. Analyses on patients with a prescription from their GP and complete original data (n = 100). Predictors of improvement in OABq-SF.** NI = Not included; *NI due to multicollinearity.
(DOCX)

## Acknowledgments

We are grateful for the assistance of Marjan Roelofs in collecting the data. We also thank Dr Robert Sykes (www.doctored.org.uk) for providing editorial services

## Author Contributions

**Conceptualization:** Martijn G. Steffens, Marco H. Blanker.

**Data curation:** Henk van der Worp.

**Formal analysis:** Henk van der Worp, Boudewijn J. Kollen.

**Investigation:** Tom Vermist.

**Methodology:** Marco H. Blanker.

**Project administration:** Marco H. Blanker.

**Supervision:** Marco H. Blanker.

**Writing – original draft:** Henk van der Worp.

**Writing – review & editing:** Henk van der Worp, Boudewijn J. Kollen, Tom Vermist, Martijn G. Steffens, Marco H. Blanker.

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
