## [Decision Letter · Decision Letter 0]

3 Jul 2019

PONE-D-19-17487

Symptom improvement and predictors associated with improvement after 6 weeks of alpha-blocker therapy: An exploratory, single-arm, open-label cohort study

PLOS ONE

Dear Dr van der Worp,

Thank you for submitting your manuscript to PLOS ONE. After careful consideration, we feel that it has merit but does not fully meet PLOS ONE’s publication criteria as it currently stands. Therefore, we invite you to submit a revised version of the manuscript that addresses the points raised during the review process.

A number of important but minor concerns were raised by the reviewer that requires clarification and a detailed response.

We would appreciate receiving your revised manuscript by Aug 17 2019 11:59PM. To enhance the reproducibility of your results, we recommend that if applicable you deposit your laboratory protocols in protocols.io, where a protocol can be assigned its own identifier (DOI) such that it can be cited independently in the future. For instructions see: http://journals.plos.org/plosone/s/submission-guidelines#loc-laboratory-protocols

We look forward to receiving your revised manuscript.

Kind regards,

Praveen Thumbikat

Academic Editor

PLOS ONE

Journal Requirements:

2. Please include additional information regarding the survey or questionnaire used in the study and ensure that you have provided sufficient details that others could replicate the analyses. For instance, if you developed a questionnaire as part of this study and it is not under a copyright more restrictive than CC-BY, please include a copy, in both the original language and English, as Supporting Information.  Moreover, please define the follow-up period for this study.

Reviewers' comments:

Reviewer's Responses to Questions

**Comments to the Author**

1. Is the manuscript technically sound, and do the data support the conclusions?

Reviewer #1: Yes

2. Has the statistical analysis been performed appropriately and rigorously? 

Reviewer #1: Yes

3. Have the authors made all data underlying the findings in their manuscript fully available?

Reviewer #1: No

4. Is the manuscript presented in an intelligible fashion and written in standard English?

Reviewer #1: Yes

5. Review Comments to the Author

Reviewer #1: Authors describe the findings from a single arm, open label clinical study on alpha blocker for male LUTS. The finding of greater benefit from alpha-blocker in participants with more severe symptoms at baseline is highly plausible.

Following concerns were noted

1) Need to specify the medications that were consumed by participants but deemed to not have an effect on LUTS by the authors. How many such medications were used by patients

2) Considering the 3.89 OR for PGI improvement with 6 medications, it is likely that the assumption of medications not having an effect on LUTS is untrue or the medications are affecting urine production.

3)Table 3 and 4 have negative values as predictors, does the negative sign imply that the variable worsens the LUTS

4) scores on (i.e., the IPSS and OAB-q SF) are subjective and not objective measures of improvement

6. PLOS authors have the option to publish the peer review history of their article (what does this mean?). If published, this will include your full peer review and any attached files.

Reviewer #1: Yes: Pradeep Tyagi

---

## [Author Response · Author response to Decision Letter 0]

12 Jul 2019

Reply to editor comments:

>>We noted some errors in the manuscript style (capitals in title) and corrected those (Page 3, line 57 and page 9, line 150)

2. Please include additional information regarding the survey or questionnaire used in the study and ensure that you have provided sufficient details that others could replicate the analyses. For instance, if you developed a questionnaire as part of this study and it is not under a copyright more restrictive than CC-BY, please include a copy, in both the original language and English, as Supporting Information. Moreover, please define the follow-up period for this study.

>>For this study we used the Dutch versions of the IPSS and the OAB-q that have been referenced in the manuscript. We did not develop a questionnaire but only asked for some demographic and disease specific information, in addition to aforementioned questionnaires. These questions have been described in the manuscript (page 4, lines 73-74 and 83-85). As the questions were in Dutch, we doubt if we needed to make the complete questionnaire available.

3. We note that you have indicated that data from this study are available upon request. PLOS only allows data to be available upon request if there are legal or ethical restrictions on sharing data publicly. If there are no restrictions, please upload the minimal anonymized data set necessary to replicate your study findings as either Supporting Information files or to a stable, public repository.

>>There are no legal or ethical restrictions on sharing data. We have now added a minimal anonymized dataset as a supporting Information file.

 

Reply to reviewer comments:

1) Need to specify the medications that were consumed by participants but deemed to not have an effect on LUTS by the authors. How many such medications were used by patients.

>>We made a frequency table of the other medications that were used. We presented the medications that were used by ≥5% percent of the participants in a supplementary file (S1 Table). And we referred to this table in the results section:

Page 6, line 135: An overview of the medication that was used by the patients is given in supporting file S1 Table.

2) Considering the 3.89 OR for PGI improvement with 6 medications, it is likely that the assumption of medications not having an effect on LUTS is untrue or the medications are affecting urine production.

>>We agree that it might be possible that other medications could influence LUTS or urine production, but we do not feel that this is very likely. First of all and most important, the other medication was initiated in the period preceding baseline assessment and had not changed during the observation period. Therefore it is not likely that symptoms improved during follow-up because of that medication. Secondly, we controlled for medication that is known to have an effect on urinary symptoms and we did not find a significant association between this variable and the outcomes. Because of this we feel it is unlikely that other medication affected outcomes. We therefore sought an explanation for the relation between the use of multiple types of medication and improvement of symptoms elsewhere, and found several studies that have shown a link between the use of multiple types of medication and adherence. For transparency we have provided a list of the most used types of medication (S1 Table). We added the following to the Discussion:

Page 17, lines 221-226: Another explanation is that other types of medication lead to improvement of LUTS, however this is less likely. First of all, other medication was initiated before baseline, and remained unchanged during the observation period. Secondly, in our analyses we looked at known specific drugs with the potential to affect LUTS and found no relation with change in symptoms, although we cannot rule out that there are other types of drugs that have an influence on LUTS.

3)Table 3 and 4 have negative values as predictors, does the negative sign imply that the variable worsens the LUTS

>>A lower score on the IPSS means less symptoms and therefore a negative sign in the regression coefficients indicates an improvement of symptoms. To emphasize this we have added this information to the Materials en Methods section and to the footnotes of the two tables.

Page 6, line 127: Because a higher IPSS score indicates more symptoms, regression coefficients of improvement will have a negative sign.

Table footnote: Negative regression coefficients indicate improvement of symptoms.

4) scores on (i.e., the IPSS and OAB-q SF) are subjective and not objective measures of improvement

>>We agree that both the IPSS and OAB-q SF are not objective measures of symptoms. The point we tried to make is that both questionnaires are more objective than the PGI-I because they ask for measurable events such as the frequency of urination, whereas the PGI-I asks for the more qualitative impression of improvement. We now replaced the term ‘objective’ with ‘quantitative’ to make this more clear.

Page 16, line 213: However, no such relationship existed between current alpha-blocker use and the more quantitative measures of improvement (i.e., the IPSS and OAB-q SF), suggesting that any observed improvements in symptoms could not be attributed to changes in either the frequency of urination or episodes of incontinence.

---

## [Decision Letter · Decision Letter 1]

16 Jul 2019

Symptom improvement and predictors associated with improvement after 6 weeks of alpha-blocker therapy: An exploratory, single-arm, open-label cohort study

PONE-D-19-17487R1

Dear Dr. van der Worp,

We are pleased to inform you that your manuscript has been judged scientifically suitable for publication and will be formally accepted for publication once it complies with all outstanding technical requirements.

With kind regards,

Praveen Thumbikat

Section Editor

PLOS ONE

Additional Editor Comments (optional):

Reviewers' comments:

Reviewer's Responses to Questions

**Comments to the Author**

1. If the authors have adequately addressed your comments raised in a previous round of review and you feel that this manuscript is now acceptable for publication, you may indicate that here to bypass the “Comments to the Author” section, enter your conflict of interest statement in the “Confidential to Editor” section, and submit your "Accept" recommendation.

Reviewer #1: All comments have been addressed

2. Is the manuscript technically sound, and do the data support the conclusions?

Reviewer #1: Yes

3. Has the statistical analysis been performed appropriately and rigorously? 

Reviewer #1: Yes

4. Have the authors made all data underlying the findings in their manuscript fully available?

Reviewer #1: Yes

5. Is the manuscript presented in an intelligible fashion and written in standard English?

Reviewer #1: Yes

6. Review Comments to the Author

Reviewer #1: (No Response)

7. PLOS authors have the option to publish the peer review history of their article (what does this mean?). If published, this will include your full peer review and any attached files.

Reviewer #1: Yes: Pradeep Tyagi

---

## [Editor Report · Acceptance letter]

17 Jul 2019

PONE-D-19-17487R1

Symptom improvement and predictors associated with improvement after 6 weeks of alpha-blocker therapy: An exploratory, single-arm, open-label cohort study

Dear Dr. van der Worp:

I am pleased to inform you that your manuscript has been deemed suitable for publication in PLOS ONE. Congratulations! Your manuscript is now with our production department.

With kind regards,

on behalf of

Dr. Praveen Thumbikat

Section Editor

PLOS ONE